# Characterization of Antimicrobial Poly(Lactic Acid)- and Polyurethane-Based Materials Enduring Closed-Loop Recycling with Applications in Space

**DOI:** 10.3390/polym16050626

**Published:** 2024-02-25

**Authors:** Andrew J. D’Ovidio, Brian Knarr, Alexander J. Blanchard, Gregory W. Bennett, William Leiva, Bin Duan, Jorge M. Zuniga

**Affiliations:** 1Department of Biomechanics, University of Nebraska at Omaha (UNO), Omaha, NE 68182, USA; adovidio@unomaha.edu (A.J.D.); bknarr@unomaha.edu (B.K.); 2National Aeronautics and Space Administration (NASA), Huntsville, AL 35808 USA; alexander.j.blanchard@nasa.gov; 3Department of Adult Restorative Dentistry, University of Nebraska Medical Center (UNMC), Omaha, NE 68198, USA; greg.bennett@unmc.edu (G.W.B.); bin.duan@unmc.edu (B.D.); 4Bucharest University of Economic Studies, 010374 Bucharest, Romania; leivawilliam21@stud.ase.ro

**Keywords:** additive manufacturing, closed loop, recycling, 3D printing, poly(lactic acid), polyurethane, antimicrobial, tensile, compression

## Abstract

Recent studies have shown that astronauts experience altered immune response behavior during spaceflight, resulting in heightened susceptibility to illness. Resources and resupply shuttles will become scarcer with longer duration spaceflight, limiting access to potentially necessary medical treatment and facilities. Thus, there is a need for preventative health countermeasures that can exploit in situ resource utilization technologies during spaceflight, such as additive manufacturing (i.e., 3D printing). The purpose of the current study was to test and validate recyclable antimicrobial materials compatible with additive manufacturing. Antimicrobial poly(lactic acid)- and polyurethane-based materials compatible with 3D printing were assessed for antimicrobial, mechanical, and chemical characteristics before and after one closed-loop recycling cycle. Our results show high biocidal efficacy (>90%) of both poly(lactic acid) and polyurethane materials while retaining efficacy post recycling, except for recycled-state polyurethane which dropped from 98.91% to 0% efficacy post 1-year accelerated aging. Significant differences in tensile and compression characteristics were observed post recycling, although no significant changes to functional chemical groups were found. Proof-of-concept medical devices developed show the potential for the on-demand manufacturing and recyclability of typically single-use medical devices using antimicrobial materials that could serve as preventative health countermeasures for immunocompromised populations, such as astronauts during spaceflight.

## 1. Introduction

### 1.1. The Role of Antimicrobial Polymers for In-Space Manufacturing

In 2014, 28 astronauts aboard the International Space Station (ISS) exhibited altered immune responses during their six-month spaceflight mission [1]. The cause of altered immune response behavior remains unidentified, but is potentially attributable to the austere environmental conditions of space and spaceflight, such as radiation, microbes, stress, microgravity, altered sleep cycles, and isolation [1,2,3]. The resulting alterations in immune response leave astronauts susceptible to illness that limit longer duration space exploration including those identified as goals in the National Aeronautics and Space Administration’s (NASA’s) 2022 strategic plan [1,2,4,5,6]. Spaceflight has been found to contribute to ‘asymptomatic viral shedding’, where no clinical signs or symptoms are noticed [2]. The absence of clinical signs or symptoms further delays a treatment timeline that is already costly from afar, jeopardizing astronaut safety for longer spaceflight missions. Furthermore, *methicillin-resistant Staphylococcus aureus* in microgravity-analog conditions has exhibited enhanced antibiotic resistance, in turn suggesting decreased efficacy of antibiotic treatment options during spaceflight [7]. Space exploration beyond the ISS, such as lunar or planetary destinations, will inherently bring new challenges to ensuring the health and safety of astronauts as medical conditions will arise and potentially pose greater long-term health risks due to their altered immune behavior. Thus, there is a critical need for preventative health countermeasures to reduce immune risks during spaceflight and facilitate future long-duration missions.

Additive manufacturing is suggested to be a suitable method for on-demand manufacturing of medical devices in a zero-gravity environment that enhances the health and safety of crewmembers [4,8,9,10,11,12]. However, the immune dysregulation exhibited by astronauts during spaceflight, in conjunction with the potentially enhanced antibiotic resistance and virulence, suggests limitations in its utilization [4]. It is critical that medical devices remain deficient of viruses and bacteria, especially prior to reuse [4]. The 2019 ISS maintenance logistical model approximates 3190 kg needs to be upmassed annually for corrective and preventative spares, 95% of which is typically never used [11]. This logistical model means that over 3000 kg of spare parts and material is wasted, which is not feasible for long-duration spaceflight and jeopardizes crew safety. Medical devices contribute not only to the increasing use of plastics, but specifically the increase in single-use plastics commonly used for either manufacturing single-use medical devices, such as urinary collection devices (UCDs) and otoscope speculums used on the ISS, or device packaging [13,14]. NASA’s logistical analysis shows that in-space manufacturing compatible with recycling of a common raw material can contribute to a closed-loop, reconfigurable or adaptable system that allows crewmembers to manufacture on-demand spares in order to adapt to their evolving needs and unforeseen scenarios [11]. Thus, there is a critical need for the development of antimicrobial, recyclable materials for the manufacturing of medical devices associated with bacterial or viral growth [1,2,4,7].

### 1.2. The Use of Copper Additives for the Development of Antimicrobial Materials

In 2015, a study from the *International Journal of Molecular Sciences* suggested the addition of copper nanoparticles within a polymer matrix can exhibit biocidal effects at low concentrations [15]. The resulting antimicrobial polymer has promising applications for the development and production of medical devices associated with bacterial risk [15]. Doremalen et al. suggested that materials common in medical settings, such as stainless steel and polypropylene, promote COVID-19 viability [16]. Using Bayesian regression modeling, the authors reported median half-life reductions in COVID-19 to occur after 5.63 h (C.I. = 4.59–6.86) and 6.81 h (C.I. = 5.62–8.17) when exposed to stainless steel and polypropylene surfaces, respectively. In comparison, copper surfaces reported a median half-life reduction occurring at 0.774 h (C.I. = 0.427–1.19) with no viable COVID-19 virus detected after 4 h of exposure. Both stainless steel and polypropylene retained the viability of COVID-19 virus for up to 72 h following exposure, suggesting copper surfaces exhibit greater effectiveness in reducing viral viability than standard stainless steel and polypropylene materials [16].

Previous investigations further support the findings of Doremalen et al. in evaluating viral deactivation of textiles infused with copper oxide particles [17,18,19,20]. A 2007 study in *Antimicrobial Agents and Chemotherapy* examined the effects of copper oxide-infused textile filters against several viruses including yellow fever, influenza A virus, measles, respiratory syncytial, parainfluenza 3, HIV-1, adenovirus type 1, and cytomegalovirus [18]. Results show that filters containing copper oxide particles significantly reduced viral titers across all inoculations tested [18]. Furthermore, USA Institute for Occupation Health and Safety N95 respiratory protective face masks were infused with copper oxide particles and tested against human influenza A (H1N1) and avian influenza (H9N2) [17]. Borkow et al. report highly effective anti-influenza properties against H1N1 and H9N2 viruses without altering the face mask’s physical protective barriers [17]. These results suggest a wide range of applications for copper-based antimicrobial materials in neutralizing and reducing infectious viral loads [15,16,17,18,21].

The purpose of the current study was to test and validate recyclable antimicrobial materials compatible with additive manufacturing and the effect of a closed-loop recycling system on antimicrobial, mechanical, and chemical properties. In addition, we aimed to develop proof-of-concept medical devices typically associated with being single-use to identify the feasibility of producing recyclable, sterile medical devices. We hypothesized that both poly(lactic acid)- and polyurethane-based materials containing a copper-based composite additive to confer biocidal activity would express robust antimicrobial efficacy and longevity in viral load reduction before and after a single closed-loop recycling cycle due to the inorganic nature of the antimicrobial additive and mechanical mechanism of recycling. The results of the current study may provide a foundation for the future evaluation of recyclable and antimicrobial materials for in situ resource utilization during spaceflight, as well as reduce material waste.

## 2. Materials and Methods

### 2.1. Antimicrobial Material Development

The antimicrobial poly(lactic acid)- and polyurethane-based materials used were developed (Copper3D, Santiago, Chile) using high-quality base polymer pellets supplemented with a copper-based composite additive to confer biocidal properties [22]. Both materials contain < 1% of their weight in additive. Copper was the primary active ingredient used to formulate the additive accounting for 22% of the additive’s composition. The additive was physically embedded within the polymer and used zeolites as a carrier to enhance antimicrobial activity.

### 2.2. Additive Manufacturing Specifications

The Redwire Space Additive Manufacturing Facility is the only permanent commercial platform currently operating in orbit. Comparable commercial 3D printers (Raise 3D Pro2 Plus, Raise 3D Technologies Inc., Irvine, CA, USA) were utilized for all manufacturing of poly(lactic acid)-based and polyurethane-based test samples. Samples were manufactured at 40% infill density (hexagon pattern), 50 mm/s print speed, 150–200 mm/s travel speed, printing temperature of 200 °C, 0.15 mm layer height, 1 mm shell thickness, and on a 50 °C heated bed. The Raise 3D Pro2 Plus printer has capabilities comparable to the Additive Manufacturing Facility with 0.01 mm to 0.25 mm layer resolution, 30 mm–150 mm/s printing speed, 300 °C maximum operating nozzle temperature, and a large build volume (305 mm × 305 mm × 605 mm).

### 2.3. Recycler Specifications

The Redwire Space Recycler is a polymer recycling unit aboard the ISS capable of recycling materials into filament to be reprinted with the Redwire Space Additive Manufacturing Facility. Comparable on-Earth systems were used for 1 closed-loop recycling cycle. A desktop extruder (Precision 450, 3devo B.V., Utrecht, The Netherlands) and a shredder (Reclaimer, Filabot, VT, USA) were used to recycle poly(lactic acid)-based and polyurethane-based materials for manufacturing test samples. The extruder is equipped with a high-end tempered precision screw and automatic neat spooling that offer comparable compatibilities to the Redwire Space Recycler. After shredding, the particulates were placed into the desktop extruder and re-extruded to a 1.75 mm filament diameter to be reprinted. The desktop extruder consists of three separate heating zones (feeding, transitioning, and metering) and four independently controlled heating units (H4-H1) to facilitate gradual melting as the material travels down the barrel for re-extrusion and spooling. The poly(lactic acid)-based material was extruded at 3.0 rpm using a bell-shaped barrel temperature parameter design: 168 °C (H4), 175 °C (H3), 175 °C (H2), and 170 °C (H1) from feeding to metering zones, respectively. The polyurethane-based material was extruded at 6.0 rpm using a gradually increasing barrel temperature profile: 195 °C (H4), 200 °C (H3), 204 °C (H2), and 208 °C (H1) from feeding to metering zones, respectively.

### 2.4. Antimicrobial Effectiveness and Longevity Testing (ISO 22196 [23])

Nine flat test samples (5 cm × 5 cm × 1 cm) per thermoplastic material were manufactured and tested before and after 1 closed-loop recycling cycle (36 samples total). Antimicrobial effectiveness and longevity of the materials were tested by an independent laboratory (Situ Biosciences LLC, wheeling, Chicago, IL, USA) following standard procedures for ISO 22196 before and after applying a heat-based accelerating aging agent for a period of 1 month, equivalent to a 1-year period of aging. Antimicrobial longevity is tested with a heat-based accelerating aging agent (55 °C) using a standard oven heat-accelerated aging protocol for standard environmental conditions. Antimicrobial effectiveness using ISO 22196 has been widely applied to test the antimicrobial activity of copper composites in medical devices [22]. The ISO 22196 is designed to measure the antimicrobial properties of a solid plastic surface incubated with bacteria. *Methicillin-resistant Staphylococcus aureus* was tested because it is a known leading cause of a variety of infections acquired within the home and hospital settings, and exhibited enhanced antibiotic resistance in microgravity-analogue conditions [7]. The basis of this test is the incubation of the bacterial inoculum in contact with the poly(lactic acid)-based and polyurethane-based polymer materials for a 24 h period. The inoculated bacteria are recovered, and the concentration of the organism is determined. Performance was determined by comparison of the recovered organism incubated in a control material with the 3D-printed antibacterial composite before and after a 24 h incubation period. The bacterial count was then standardized to 5 log10 colony-forming unit (CFU)/mL by measuring the optical density of the bacteria at 600 nm and they were kept at 37 °C for 2 h to avoid the lag phase of the kinetic bacterial growth. The measurements were corrected with the dilution factor that divides the numbers of CFUs by the product of the dilution factor and the volume of the plated diluted suspension to determine the total number of bacteria within the original solution. All the experiments were repeated at least three times on different days. The samples, tools, glass materials, and wells were sterilized in an Orthmann autoclave for 20 min at 1 bar and 140 °C before the experiment.

### 2.5. Contamination Mitigation Procedures for Antimicrobial Testing Samples

To minimize the risk of external microbial contamination at the point of manufacturing and shipping for independent laboratory testing, the following procedures were developed in-house. All native state filaments remained in their initial packaging until needed for manufacturing. Since contact is the primary source of contamination for poly(lactic acid) and polyurethane materials, all potential sources of contact during the printing and recycling processes were disinfected using 70% isopropyl alcohol. This includes, but is not limited to, the filament feeding systems within the printer, the build platform, the shredding blades, and the material storage containers, among others. Printing temperatures for both materials were high enough to eliminate any potential contamination as material was extruded. Samples were disinfected with isopropyl alcohol and stored in a sealed plastic bag for shipping to be tested. Laboratory procedures were also in place to mitigate contamination. All samples, tools, glass materials, and wells were sterilized in an Orthmann autoclave for 20 min at 1 bar and 140 °C before the experiment.

### 2.6. Tensile Testing to Assess Ultimate Strength, Elastic Modulus, and Elongation to Failure (ASTM D638 [24])

Nominal dimensions for tensile specimens were based on type IV “bowtie” specimens in ASTM D638 (total length = 115 mm; width = 19 mm; thickness = 4 mm). Nine specimens were manufactured per poly(lactic acid)-based and polyurethane-based material in the native state and after 1 recycling cycle (18 total specimens). Tensile specimens were preloaded to a minimum of 1 lbf at 0.05 in/min, then pulled at a failure rate of 0.2 in/min. All equipment used for these tests was calibrated as per applicable ASTM standards.

### 2.7. Compression Testing to Measure Compressive Strength and Compressive Modulus (ASTM D695 [25])

Nine compression tests specimens (diameter = 12.7 mm, length = 25.4 mm) were manufactured per poly(lactic acid)-based and polyurethane-based material in the native state and after 1 recycling cycle (18 total specimens). Specimens were placed between the platens of an ASTM D695-compliant compression fixture and compressed at a rate of 0.050 in/min, as per industry standards. The specimens were tested until reaching a local maximum or at least 80% compressive strain, whichever occurred first.

### 2.8. Fourier-Transform Infrared Spectroscopy (FTIR)

Nine specimens were manufactured per poly(lactic acid)-based and polyurethane-based material in the native state and after 1 recycling cycle (18 total specimens). FTIR spectroscopy was used to assess the presence of functional chemical groups within the specimens [11]. Through this, FTIR spectroscopy assesses potential feedstock material chemical changes due to aging or environmental exposure (i.e., humidity, radiation) [11].

### 2.9. Development of Medical Devices Prototype

A 4 mm × 5 mm adult size Gruber-style oval ear speculum was manufactured using the poly(lactic acid)-based material. A male and female UCD adapter (funnel) was manufactured using the polyurethane-based material. Materials were selected based on the functionality of the device and its proof-of-concept nature. Two UCDs were designed to be compatible with the adapter tubing, the disconnect hardware attached to the storage bag, and the Universal Waste Management System [14]. The UCD designs consider use during extravehicular activity operations outside the spacecraft. All CAD models were designed and scaled using Autodesk Fusion 360 (Fusion 360, Autodesk Inc., San Rafael, CA, USA).

### 2.10. Statistical Analysis

Several MANOVAs were conducted separated by material and mechanical characterization method (i.e., tensile and compression) to evaluate the effects of a single closed-loop recycling cycle (i.e., recycling state) on mechanical properties. Prior to running the MANOVAs, Shapiro–Wilk’s and Box’s M tests were conducted to evaluate multivariate normality and homogeneity, respectively. Univariate tests were performed following multivariate analysis if a significant effect was observed. For all comparisons, an alpha level equal to or higher than 0.05 was considered statistically significant.

## 3. Results

### 3.1. Antimicrobial Effectiveness and Longevity of Recycled Polymers

Native-state poly(lactic acid) shows high biocidal activity in reducing *methicillin-resistant Staphylococcus aureus* viral load, resulting in upwards of 99.99% efficacy pre aging and 90.86% efficacy post aging. Native-state polyurethane expresses 99.95% and 99.99% efficacy pre and post aging, respectively. Following a single closed-loop recycling cycle the poly(lactic acid) material (i.e., recycled-state) retained its biocidal properties, effectively reducing 99.99% viral load both before and after aging. However, recycled-state polyurethane exhibited a drastic decrease in efficacy from 98.91% viral load reduction pre aging to 0.00% post aging (Figure 1).

### 3.2. Tensile Testing

MANOVA analyses investigating the effect of recycling (i.e., native vs. recycled state) on tensile characteristics found the effect to be statistically significant for both the poly(lactic acid)- (Wilks’ λ = 0.071; F(3,14) = 60.779; *p* > 0.001; partial η^2^ = 0.929) and polyurethane-based materials (Wilks λ = 0.119; F(3,14) = 34.545; *p* > 0.001; partial η^2^ = 0.881). Follow-up univariate analyses were conducted to investigate the specific effect of recycling on tensile ultimate strength, elastic modulus, and elongation to failure. Univariate tests of the poly(lactic acid) reveal a significant effect of recycling on elongation failure (F(1,16) = 8.564; *p* = 0.010; partial η^2^ = 0.349). No differences were observed in poly(lactic acid)’s ultimate strength (F(1,16) = 0.157; *p* = 0.697; partial η^2^ = 0.010) or elastic modulus (F(1,16) = 2.610; *p* = 0.126; partial η^2^ = 0.140). However, the univariate tests for polyurethane revealed a significant effect on all tensile properties including ultimate strength (F(1,16) = 20.448; *p* > 0.001; partial η^2^ = 0.561), elastic modulus (F(1,16) = 81.526; *p* > 0.001; partial η^2^ = 0.836), and elongation to failure (F(1,16) = 30.640; *p* > 0.001; partial η^2^ = 0.657). Descriptive differences in tensile property trends for the poly(lactic acid)- and polyurethane-based materials are reported (Table 1). In addition, significance was found in Box’s test of equality of covariance matrices for poly(lactic acid) and Levene’s test of equality of error variances (i.e., homogeneity) in all tensile measurements of both materials. Furthermore, Shapiro–Wilk’s test for normality found significance in native-state poly(lactic acid)’s ultimate strength and elastic modulus, as well as native-state polyurethane’s elastic modulus groups.

### 3.3. Compression Testing

MANOVAs assessing the effect recycling has on compression properties found a significant effect on both the poly(lactic acid)- (Wilks’ λ = 0.026; F(2,15) = 282.021; *p* > 0.001; partial η^2^ = 0.974) and polyurethane-based materials (Wilks’ λ = 0.087; F(2,15) = 282.021; *p* > 0.001; partial η^2^ = 0.913). Univariate follow-up analysis of poly(lactic acid) reveals a significant effect of recycling on compressive strength (F(1,16) = 28.579; *p* > 0.001; partial η^2^ = 0.641) and compressive modulus (F(1,16) = 13.050; *p* = 0.002; partial η^2^ = 0.449). A significant effect of recycling was also found on polyurethane’s compressive strength (F(1,16) = 144.581; *p* > 0.001; partial η^2^ = 0.900). All of the significant effects of recycling found on poly(lactic acid)’s and polyurethane’s compression characteristics suggest large effect sizes. Descriptive differences in compressive property trends for the poly(lactic acid)- and polyurethane-based materials are reported (Table 2). Additional significance was found in Box’s test of equality of covariance matrices for both materials, as well as Levene’s test of equality of error variances (i.e., homogeneity) in all compression measurements except polyurethane’s compressive modulus. Furthermore, Shapiro–Wilk’s test for normality found significance in native-state poly(lactic acid)’s compressive strength and compressive modulus.

### 3.4. Fourier-Transform Infrared Spectroscopy

The poly(lactic acid)- and polyurethane-based materials were assessed for the presence of functional chemical groups, as well as potential feedstock chemical changes, using FTIR spectroscopy. Upon inspection of the infrared spectrum fingerprint region (1500 cm^−1^–400 cm^−1^), minor shifts can be observed in both poly(lactic acid)’s and polyurethane’s absorbance peaks following one closed-loop recycling cycle (Figure 2). Furthermore, some C-H stretching over time is present in both recycled-state materials (3300 cm^−1^–2800 cm^−1^). These observations are not universal across recycled-state materials and limited to a minority of the samples tested.

### 3.5. Medical Device Prototypes

A 4 mm × 5 mm adult-size Gruber-style oval ear speculum prototype was designed using commercially available CAD modeling software (Fusion 360 version 2.0.17453, Autodesk Inc., San Rafael, CA, USA). The design was adapted to conform with the attachment point of the otoscope and provide a friction fit that reduces the likelihood of falling loose during use (Figure 3A). Poly(lactic acid) was used to manufacture the device because of its resemblance (e.g., rigid properties) to those of commercially available devices. In addition, a UCD prototype was designed based on inspiration from previous devices used by NASA [14]. Our prototype was designed to be unisex-compatible while considering the adapter disconnect hardware and dimensional constraints of the manufacturing capabilities aboard the ISS [26]. Specifically, the device features similar contact geometry to the NASA “one-size fits all” female adapter and an extended flange intended to be folded over for male users, similar to the NASA Bellows condom catheter, increasing the surface area that would reduce the likelihood of detachment during use (Figure 3B).

## 4. Discussion

The purpose of the current study was to assess the effects of closed-loop recycling on the antimicrobial, mechanical, and chemical characteristics of thermoplastic poly(lactic acid)- and polyurethane-based materials. Our main findings suggest that the antimicrobial properties of poly(lactic acid) and polyurethane materials are robust to closed-loop mechanical recycling, although a decrease in mechanical properties was observed similar to those found in previous studies [27,28]. Recycled-state poly(lactic acid) and polyurethane materials retain printability without the need for parameter adjustments, conveying capability and feasibility of manufacturing and recycling typically single-use medical devices, such as the otoscope ear speculum and UCD prototypes developed, while maintaining highly effective antimicrobial properties.

### 4.1. Changes in Antimicrobial Effectiveness and Longevity Influenced by Recycling

Based on the results from native- and recycled-state antimicrobial analyses for the poly(lactic acid)- and polyurethane-based materials, we failed to reject our hypothesis that closed-loop recycling does not significantly affect antimicrobial properties. All analyses show high biocidal activity (>90%) effectively reducing *methicillin-resistant Staphylococcus aureus* viral load, except for recycled-state polyurethane reporting 0% efficacy post aging. Native-state poly(lactic acid)’s pre- and post-aging results coincide with findings previously published by our team evaluating a poly(lactic acid)-based material against standard and *methicillin-resistant Staphylococcus aureus* and *Escherichia coli.* Thus, the validity of our native-state results is supported. Our results further suggest robust properties of the copper-based composite additive, observing high antimicrobial efficacy and long-term retention of the antimicrobial poly(lactic acid)-based material after being recycled in a closed-loop system (upwards of 99.99% viral load reduction). However, a drastic reduction in antimicrobial efficacy for recycled-state polyurethane was not expected, especially given its high biocidal expression pre aging (98.91%). It is unclear why both recycled-state materials are highly effective pre aging, but only polyurethane decreased following the aging process. However, it could be speculated that either (i) the aging process (i.e., constant exposure to 55 °C for 28 d) induces a material-altering event affecting the polyurethane polymer base or copper additive, or (ii) external contamination occurred resulting in a 0% average reduction across all samples. The latter derived hypothesis is unlikely given the disinfection protocols in place from test sample manufacturing to laboratory receipt of samples, as well as the sterilization procedures for all equipment and samples established at the testing laboratory.

The poly(lactic acid)- [13,22,29] and polyurethane-based materials evaluated in the current study are derivates of organic materials, such as corn- and petroleum-based products, respectively. However, the copper-based composite additive suspended in both polymer matrices is in an inorganic state. While further analysis would need to be conducted on recycled polyurethane post aging, it is plausible that the aging process could have induced changes to the polyurethane or the copper composite additive itself by exposure to low, constant heat that cannot be measured using FTIR analysis. Additional complexities arise when considering the effect of constant low heat exposure on the recycled polymer because of the organic derivation of the base polymer and addition of an inorganic copper additive. The previous literature has reported the sensitivity of organic compounds to heat and processing temperatures [28,30,31]. Inorganic antimicrobial additives have an advantage over their organic counterparts with regard to thermal stability and resistance to heat, but are at the detriment of the stability of their release mechanism (i.e., rate of ion release), which could result in rapid oxidation and a loss of antimicrobial efficacy [31]. However, a counterargument could be made as to why the effect of recycling polyurethane, with processing temperatures reaching upwards of 210 °C, did not affect pre-aging antimicrobial efficacy. It can be speculated that constant low heat exposure affects recycled polyurethane’s physical properties as a copper ion transporter [32,33], rather than inducing chemical changes in the material or its copper additive. Special considerations should be made for how aging affects the contact surface area within recycled-state polyurethane samples due its importance in the rapid, yet controlled, release of copper ions [32,33,34]. It is important to note that the derived hypotheses and speculations serve to rationalize potential contributors to our unexpected results obtained on recycled-state post-aging polyurethane’s antimicrobial longevity and are beyond the scope of the current study. Future investigations are required to validate the potential contributing factors.

### 4.2. Differences in Mechanical and Chemical Characteristics Influenced by Recycling

The mechanical properties of both the poly(lactic acid)- and polyurethane-based materials were significantly affected by closed-loop recycling. The elongation to failure of poly(lactic acid) was the only tensile characteristic significantly affected by recycling, decreasing in average elongation length. Comparisons of native- and recycled-state polyurethane saw significant decreases in ultimate strength and elongation to failure, as well as a significant increase in elastic modulus. Increases in elastic modulus are indicative of greater material stiffness and lower elastic strain. Our results align similarly with previous studies [27,28]; however, statistically significant differences were seen following several recycling cycles, whereas our observations arose after a single cycle. Both the poly(lactic acid)- and polyurethane-based materials saw significant increases in compressive strength as a result of recycling. Interestingly, compressive modulus was significantly affected only in the poly(lactic acid) material. In conjunction, poly(lactic acid)’s compressive modulus showed a decreasing trend contrasting with the increasing trends seen in compressive strength (Table 2). Typically, compressive modulus and compressive strength trend in parallel [28,35] as an increasing value in compressive modulus is indicative of an upward trend in material stiffness. An inverse relationship between the two measures is typically seen for tensile strength and elastic modulus. It is possible that a single recycling cycle does not significantly introduce material degradation characteristics, such as reduced molecular weight and polymer chain length, that would significantly affect compressive characteristics in a parallel manner [28,35]. However, determining degradation is beyond the scope and capabilities of the current study.

Chemical characteristics do not appear to be significantly affected by recycling. Within the fingerprint region of the infrared spectrum, minor shifts can be observed in both the poly(lactic acid) and polyurethane materials post recycling (Figure 2). While both materials also expressed C-H stretching over time in the functional chemical group region, these were not necessarily indicators of significant chemical changes. A potential indicator that significant chemical changes in the material could have occurred would be a change in functional chemical groups, such as an alkene stretch changing to a CH group over time (recycling). Care should be taken when considering FTIR spectroscopy results due to its limitations in providing only identifying information about a given material, such as the material fingerprint and presence of functional chemical groups [11]. FTIR may provide insight on potentially significant chemical changes in the materials, but further analyses, such as nuclear magnetic resonance, gel permeation chromatography, or differential scanning calorimetry, would be necessary to determine material degradation. However, our results support the possibility that recycled-state polyurethane’s antimicrobial efficacy post aging (0%) is likely attributable towards physical or structural changes induced by the aging process, rather than alterations to the chemical composition of the material.

### 4.3. Description of Medical Device Prototypes

Our primary application for recyclable antimicrobial poly(lactic acid)- and polyurethane-based materials is in the development of typically single-use medical devices associated with microbial risk, such as an otoscope ear speculum and UCDs [14]. Poly(lactic acid) was used to manufacture the otoscope ear speculum prototype because ear speculums, such as the 4 mm × 5 mm adult size Gruber-style oval ear speculum used to guide design, are traditionally manufactured in rigid materials as they will be inserted into the ear canal. Our otoscope ear speculum was adapted at the otoscope connection point to create a firm friction fit with the otoscope itself (Figure 3). Additionally, the criteria for UCDs call for a material that is biocompatible, resistant to degradation via water (e.g., urine), and flexible to prevent discomfort while wearing underneath clothing. Our UCD was modeled after the NASA “one-size fits all” female urinary collection adapter [14]. However, we sought to create a unisex UCD to allow for universal use (Figure 3).

### 4.4. Limitations

Several limitations exist within the present study and should be considered for future investigations. A major limitation of thermoplastic polymers is their affinity for water. Both poly(lactic acid) and polyurethane materials are sensitive to ambient humidity, resulting in variable extrusion success while recycling. Specifically, materials are highly susceptible to absorbing moisture while in the “regrind” form post shredding prior to filament extrusion due to their increased surface area. Polyurethane, unlike poly(lactic acid), is also susceptible to moisture absorption when in filament form to be printed. We recommend limiting the environmental exposure of reground material and filaments to mitigate moisture absorption issues, which can be accomplished using an airtight or humidity-controlled storage bin. Another major limitation is the potential violations of MANOVA assumptions, specifically Box’s M and Levene’s tests. MANOVA statistical analyses are beneficial in our case when evaluating several dependent variables for tensile and compressive characteristics, but they are highly sensitive to variability in normality and homogeneity assumptions. One contributor in the current study is sample size. Each material was used to manufacture a sample size of nine (*n* = 9) per mechanical characterization (tensile and compression) and recycling state (native and once recycled). Looking at the means of each variable assessed (Table 2), a large variation in values can be observed. Potential resolutions for this sensitivity are twofold: (i) consider alternative statistical tests less sensitive to variation within groups, such as the dependent Student’s T test or the multivariate Kruskal–Wallis test (i.e., non-parametric version of MANOVA), or (ii) calculate an appropriate sample size using power analysis and reassess assumptions for MANOVA analysis to determine whether it is still appropriate. Closed-loop mechanical recycling itself is another limitation. Practical applications of mechanical recycling typically supplement recycled material with virgin (i.e., native) feedstock to reduce decrements in mechanical properties, which affect mechanical property measurements but also impact the extrusion process to create a spool of recycled material. Our setup was limited to desktop devices that are not as precise as commercial-grade filament production lines, therefore limiting the quality control of recycled material production. However, a 100% closed-loop system was vital to maintaining antimicrobial assessment integrity. Lastly, the quality control of test samples and recycled filament spools was limited to physical measuring instruments (e.g., calipers) and infrared sensor diameter measurements built into the extruder. Future studies should consider more precise pre-testing measurements to ensure samples are reproduced with similar quality.

### 4.5. Future Directions

Our results primarily fail to reject our hypothesis that closed-loop mechanical recycling does not have a significant effect on the antimicrobial properties of thermoplastic polymers. However, further investigation into the characteristics influencing recycled-state polyurethane post aging should be considered. The FTIR analysis conducted in the current study did not provide substantial evidence that significant chemical changes in the polyurethane material occurred, at least prior to aging. Thus, it is possible that the aging process affected the material properties related to copper ion transport and release necessary for antimicrobial behavior. Further insight into how recycling or aging could affect copper ion transport and release may be provided by analyzing the leeching potential of the antimicrobial additive from native- and recycled-state materials. Future studies should consider alternative quantitative analyses to better determine changes within the material as a result of both recycling and aging processes. Scanning electron microscopy and elemental mapping using energy-dispersive X-ray spectroscopy could be used to assess the distribution of the antimicrobial additive within manufactured samples, which is currently assumed to follow a homogenous distribution throughout the recycled-state filament. In addition, additional analyses for determining degradation should be considered to supplement FTIR data. Nuclear magnetic resonance, gel permeation chromatography, or differential scanning calorimetry may be considered suitable to supplement FTIR data. Mechanical characteristics, such as strain hardening or fatigue, could be affected by the recycling process and further investigated by analyzing the characteristics of stress–strain curves. Furthermore, future studies should consider investigating differences between chemical and mechanical recycling of antimicrobial polymers.

## 5. Conclusions

In conclusion, antimicrobial poly(lactic acid)- and polyurethane-based materials are robust to closed-loop mechanical recycling, retaining highly effective biocidal properties and displaying no deterministic indicators of significant chemical changes. The poly(lactic acid) material retained greater than 90% viral load reduction in both native and recycled states, as well as pre and post aging. The polyurethane material also retained over 90% efficacy in all analyses conducted except for in its recycled state post aging, which saw a decrease to 0% from its recycled state pre aging (98.91%). A majority of the mechanical properties assessed were significantly impacted by recycling, which is likely due to them being processed in a 100% closed-loop system with no introduction of virgin materials or material compounding. Specifically, the poly(lactic acid)-based material displayed significant differences in elongation to failure, compressive strength, and compressive modulus post recycling. The polyurethane-based material showed significant differences in ultimate tensile strength, elastic modulus, elongation to failure, and compressive strength. FTIR analyses for both materials showed minor shifts in the absorbance spectrum and C-H stretching over time (i.e., cycles), but no definitive markers of significant chemical changes (e.g., alkene stretch to CH group) were observed. Our findings may assist in the future development of passive preventative health countermeasures by utilizing in situ resource manufacturing methods of recyclable and antimicrobial materials for populations with heightened immune risk, such as astronauts during spaceflight.

## Figures and Tables

**Figure 1 polymers-16-00626-f001:**
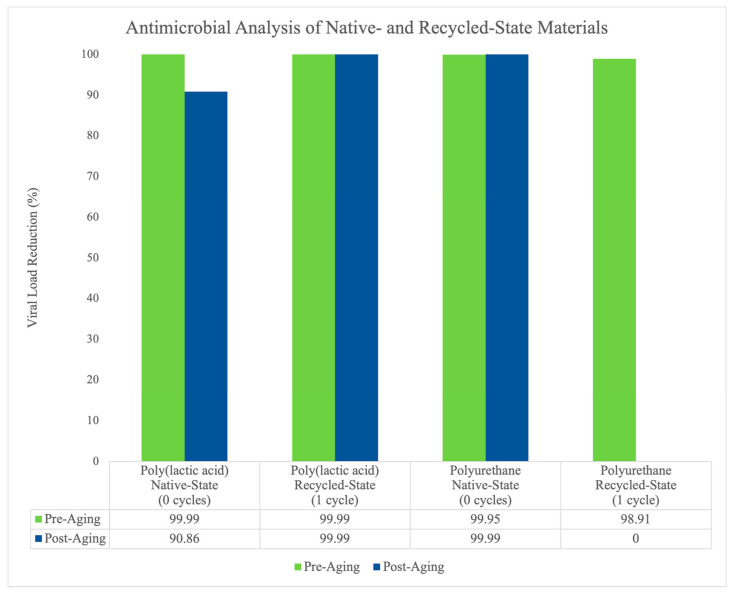
Bacterial analysis of antimicrobial efficacy and longevity for native- and recycled-state materials. Native-state poly(lactic acid) and polyurethane materials exhibit high biocidal activity, resulting in > 90% viral load reduction pre and post aging. Recycled-state polyurethane retains antimicrobial efficacy and longevity properties pre and post aging. Recycled-state polyurethane, however, shows high antimicrobial efficacy pre aging, resulting in 98.91% viral load reduction, but does not maintain longevity, decreasing to 0% viral load reduction post aging.

**Figure 2 polymers-16-00626-f002:**
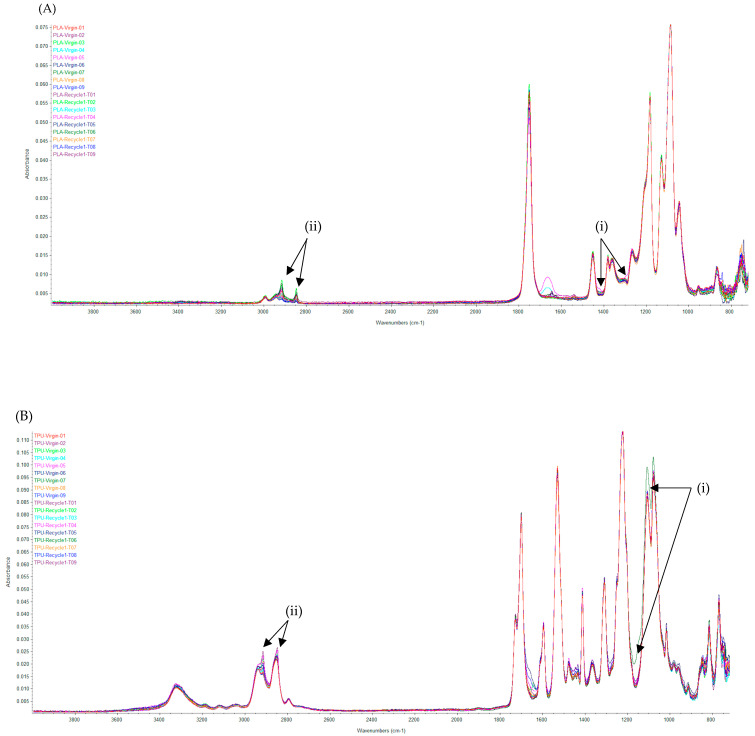
Chemical characterization via FTIR spectroscopy performed on native- and recycled-state (**A**) poly(lactic acid)- and (**B**) polyurethane-based materials. Minor shifts in absorbance peaks are observed within (i) the fingerprint region (1500 cm^−1^–400 cm^−1^), as well as (ii) C-H stretching over time in recycled-state samples from both materials.

**Figure 3 polymers-16-00626-f003:**
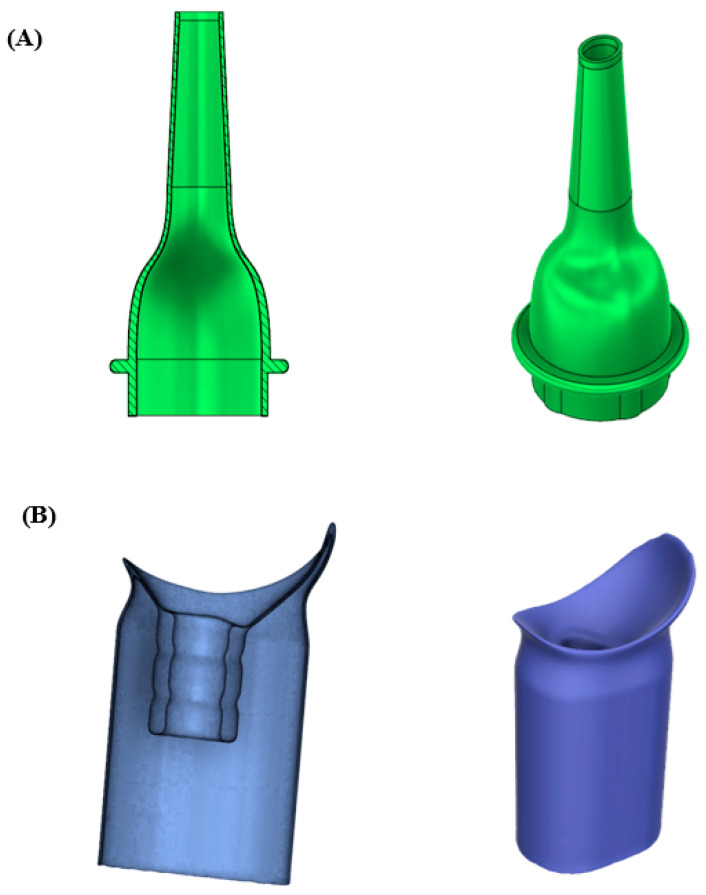
Cross-section analysis and full rendering of prototype medical devices: (**A**) prototype 4 mm × 5 mm adult size Gruber-style oval ear speculum, and (**B**) unisex urinary collection device were designed using 3D CAD modeling (Fusion 360, Autodesk Inc., San Rafael, CA, USA).

**Table 1 polymers-16-00626-t001:** Tensile properties of native and recycled materials.

Material	Recycling State	Ultimate Strength (lbf)	Elastic Modulus (psi)	Elongation to Failure (in)	Elongation %
Poly(lactic acid)	Native (0 cycles, n = 9)	201.208 (±46.755)	325,087.667 (±74,585.966)	0.050 (±0.002)	2.035 (±0.075)
	Recycled (1 cycle, n = 9)	194.739 (±14.366)	366,437.889 (±18,265.952)	0.046 (±0.002)	1.854 (±0.170)
Polyurethane	Native (0 cycles, n = 9)	127.653 (±11.907)	4,685.314 (±1,471.479)	9.339 (±0.692)	373.560 (±27.693)
	Recycled (1 cycle, n = 9)	201.21 (±46.750)	9,531.104 (±653.445)	7.587 (±0.650)	303.480 (±25.994)

**Table 2 polymers-16-00626-t002:** Compression properties of native and recycled state materials.

Material	Recycling State	Compressive Strength (lbf)	Compressive Modulus (psi)
Poly(lactic acid)	Native (0 cycles, n = 9)	999.763 (±199.557)	184,347.000 (±43,938.906)
	Recycled (1 cycle, n = 9)	1,386.409 (±85.178)	130,601.222 (±7,842.038)
Polyurethane	Native (0 cycles, n = 9)	112.541 (±6.246)	5,552.956 (±904.867)
	Recycled (1 cycle, n = 9)	243.035 (±31.953)	5,873.806 (±792.092)

## Data Availability

Data are available upon request. Please enquire with the corresponding author; jmzuniga@unomaha.edu.

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
