# Peer review of "Characterization of Antimicrobial Poly(Lactic Acid)- and Polyurethane-Based Materials Enduring Closed-Loop Recycling with Applications in Space"

_polymers, 2024, doi:10.3390/polym16050626_

Round 1

Reviewer 1 Report

Comments and Suggestions for Authors

The manuscript Characterization of Antimicrobial Polylactic acid- and Polyure-2 thane-based Materials Enduring Closed-loop Recycling with 3 Applications in Space is well written and interesting for the readers. However, some improvements must be done before publishing.

1. Introduction should be improved with newest literature data. Most of the references are from the year of 2020 or older.

2. There is a great number of abbreviations which were used only one or two times and makes the text difficult for readers. For example, NIOSH and EVA were used only one time and MRSA two times.

3. Why only methicillin-resistant Staphylococcus aureus was used for the antimicrobial testing? Will other bacteria be tested in future researches?

4. Would it be possible to recycle these materials for more than one cycle? How that would affect mechanical and antimicrobial properties?

5. Conclusions should be improved with more detailed results of antimicrobial and mechanical testing.

Reviewer 2 Report

Comments and Suggestions for Authors

The aim of the study was to develop and validate recyclable antimicrobial materials that can also be used in 3D printers. Poly(lactic acid)- and polyurethane-based materials were evaluated for antimicrobial and mechanical properties before and after one closed-loop recycling cycle. The results show high biocidal effectiveness (>90%) of both poly(lactic acid) and polyurethane materials, while maintaining effectiveness after recycling, with the exception of recycled polyurethane, where effectiveness decreased to 0% after 1 year of accelerated aging. After recycling, significant differences were observed in the tensile and compression characteristics. The developed proof-of-concept medical devices demonstrate the potential for on-demand manufacturing and recyclability of typically single-use medical devices using antimicrobial materials that could serve as preventative health countermeasures for immunocompromised populations, such as astronauts during spaceflight.

There is no mention of copper additives in the Abstract, in the Materials and Methods section or in the Conclusions. One can only guess from the Introduction that some copper-based substance was added. There is little mention in the discussion, once about a copper-based composite and several times as an additive. But whether cooper was added in a large amount (and it is a composite) or a small amount as an additive and in what form readers do not know. Authors must complete this information because it is crucial. Were only materials with additives tested? Or without any addition. Does native mean before recycling or before adding cooper?

Moreover, FTIR analysis is not an appropriate tool for determining material degradation. It would be necessary to do more NMR or even DSC analysis, not to mention GPC analysis. Authors are asked to include DSC analyses.

Detailed comments:

1. “Polylactic acid”: Please use the IUPAC nomenclature. The names of polymers whose monomers consist of two words or more are written with parentheses. Should be “Poly(lactic acid)”. Please correct throughout the text.

2. “5.63 hours”: In scientific texts, values of quantities are expressed in acceptable SI units using and symbols for units (5.63 h). Please correct throughout the text.

3. “50mm/s”, “305mm”, etc.; According SI there should be a space between the numerical value and unit symbol. Please correct throughout the text.

4. “recycling cycle he polylactic acid material”: What did "he" mean? Please correct.

5. “n=9”: There should be a space before and after the mathematical operators such as: = and so on. Please correct throughout the text.

Round 2

Reviewer 1 Report

Comments and Suggestions for Authors

Thank You for answering the questions. The manuscript is suitable for publication in present form.

Reviewer 2 Report

Comments and Suggestions for Authors

The authors did not answer the most important question: "Moreover, FTIR analysis is not an appropriate tool for determining material degradation. It would be necessary to do more NMR or even DSC analysis, not to mention GPC analysis. Authors are asked to include DSC analyses"

Also “>90%”: There should be a space before and after the mathematical operators such as: <, > and so on. Please correct throughout the text. 

Round 3

Reviewer 2 Report

Comments and Suggestions for Authors

I would like to thank the authors for correcting the manuscript, which is now suitable for publication.